# Biocomplexity and Fractality in the Search of Biomarkers of Aging and Pathology: Mitochondrial DNA Profiling of Parkinson’s Disease

**DOI:** 10.3390/ijms21051758

**Published:** 2020-03-04

**Authors:** Annamaria Zaia, Pierluigi Maponi, Martina Zannotti, Tiziana Casoli

**Affiliations:** 1Center of Innovative Models and Technology for Ageing Care, Scientific Direction, IRCCS INRCA, 60121 Ancona, Italy; martina.zannotti@studenti.unicam.it; 2School of Science and Technology, University of Camerino, 62032 Camerino (MC), Italy; pierluigi.maponi@unicam.it; 3Center for Neurobiology of Aging, IRCCS INRCA, 60121 Ancona, Italy; t.casoli@inrca.it

**Keywords:** aging, biocomplexity, chaos game representation, fractal lacunarity, mtDNA, Parkinson’s disease

## Abstract

Increasing evidence implicates mitochondrial dysfunction in the etiology of Parkinson’s disease (PD). Mitochondrial DNA (mtDNA) mutations are considered a possible cause and this mechanism might be shared with the aging process and with other age-related neurodegenerative disorders such as Alzheimer’s disease (AD). We have recently proposed a computerized method for mutated mtDNA characterization able to discriminate between AD and aging. The present study deals with mtDNA mutation-based profiling of PD. Peripheral blood mtDNA sequences from late-onset PD patients and age-matched controls were analyzed and compared to the revised Cambridge Reference Sequence (rCRS). The chaos game representation (CGR) method, modified to visualize heteroplasmic mutations, was used to display fractal properties of mtDNA sequences and fractal lacunarity analysis was applied to quantitatively characterize PD based on mtDNA mutations. Parameter β, from the hyperbola model function of our lacunarity method, was statistically different between PD and control groups when comparing mtDNA sequence frames corresponding to GenBank np 5713-9713. Our original method, based on CGR and lacunarity analysis, represents a useful tool to analyze mtDNA mutations. Lacunarity parameter β is able to characterize individual mutation profile of mitochondrial genome and could represent a promising index to discriminate between PD and aging.

## 1. Introduction

Parkinson’s disease (PD) is a neurodegenerative disorder with heavily age-dependent prevalence affecting 1% of over 65s and more than 4% of over 85s subjects worldwide [1]. Pathological characteristics of PD are a loss of dopaminergic neurons and the presence of Lewy bodies (aggregates of α-synuclein and other proteins) in the substantia nigra. Clinical manifestation of the disease is a severe motor dysfunction resulting in resting tremor, rigidity, bradykinesia, and impaired balance.

Its causes are still unknown in spite of intense research efforts over many years. Over the past two decades, scientific opinion has varied between two positions in which only environmental factors or only genetic factors were considered to be dominant [2,3,4,5,6,7]. At present, the common opinion is that PD results from a complex genetic–environmental interaction in which aging is a contributory factor [8]. Several mechanisms have been proposed to explain cell death in PD, such as oxidative stress, mitochondrial dysfunction, apoptosis, excitotoxicity, and inflammatory responses [9]. In past decades, the interest for the genetic implication in the pathogenesis of PD has been increasing. Genetic mutations have been associated only with rare forms of PD; nevertheless, the discovery of these genes has given new insight into the pathogenesis of PD. In particular, the role of abnormal protein processing has been recognized as a major mechanism of cell death in both genetic and sporadic PD as well as in other neurodegenerative disorders [10,11].

Increasing evidence implicates mitochondrial dysfunction in the etiology of neurodegenerative disorders among which is PD [12,13,14,15]. Neurodegenerative disorders progress gradually, suggesting that quantitative factors are central to the etiology of these diseases. Mitochondrial DNA (mtDNA) is the only genetic element in human cells that is present in sufficient number of copies to cause a graded decline of functions [16]. It encodes critical elements of cellular energy production, is present in thousands of copies per cell, and has a very high mutation rate. During cell replication, the occurrence of a mtDNA mutation within a cell generates a stochastically drifted mixture of mutant and normal mtDNA (heteroplasmy). Mitochondrial defects can cause organ-specific phenotypes depending on their differently rely on mitochondrial energy. The central nervous system is the most reliant on mitochondrial energy, and therefore neurodegenerative disease(s) can be a common manifestation of mitochondrial dysfunction [17,18,19,20].

The association between mtDNA mutations and PD has been the subject of considerable debate. This could be attributable to the complexity of mitochondrial genetics and the variety of classes of mtDNA mutations relevant to neurodegenerative disease(s). Three classes of mtDNA mutations are implicated: ancient maternally inherited polymorphisms [21,22,23], recent maternally inherited pathogenic mutations [24,25], and somatic mtDNA mutations [25,26]. These last mutations accumulate in post-mitotic cells with age and amplify the biochemical effects of the former two classes. Accumulation of somatic mtDNA mutations has been shown to be an important factor in the development of PD and Alzheimer’s disease (AD) as well as in aging [27]. The role of mtDNA in PD is supported by the analysis of cybrid cells, generated by the fusion of platelets (from control or patient) as mitochondria donor with recipient cells deprived of mitochondria. The common nuclear background of cybrids allows excluding the nuclear genome effect in the bioenergetics characterization. PD cybrids show depolarized mitochondria, reduced Complex I (CI) activity, increased ROS (reactive oxygen species) production and lower ATP production, most hallmarks of PD [28].

Neuronal and systemic alterations of mtDNA have also been documented in AD [29,30], suggesting a possible common pathogenic mechanism. This aspect is supported by a recent study on neuromolecular imaging that brings the etiology of Parkinson’s to the gastrointestinal tract via beta amyloid [31]. It has been previously reported that peripheral blood samples from AD patients exhibited an increased presence of variants in 89.7% of nucleotide positions (np) along the whole mtDNA sequence when compared to control subjects [30]. In a more recent study on PD patients, it has been reported that 47 out of 58 np variants showed an increased presence of non-reference alleles in PD patients [32].

We have recently proposed a method of lacunarity analysis of chaos game representation (CGR) of mtDNA sequences [33] able to discriminate between aging and AD on the basis of mtDNA mutation profiles. The method was developed taking into account the complexity of living beings and fractal properties of many anatomic and physiologic structures, among which is mtDNA [34,35,36,37,38,39]. In particular, the concept that aging can be considered as a “secondary product” of the temporal evolution of a dynamic nonlinear system [40,41,42], governed by the laws of deterministic chaos, can explain the variability observed in the senescent phenotype [33]. In addition, the concept that a complex system with a chaotic behaviour often generates fractal structures [43] highlights the usefulness of fractal analysis as a suitable tool to measure biocomplexity and its changes with aging at both functional and structural levels [44,45,46,47].

Fractal analysis can measure variations of complexity in biosystems evolving with time by following different trajectories. Individual specific genetic-environment interactions define the senescent phenotype as normal aging, pathological aging, or successful aging [33,41]. Fractal analysis, therefore, represents a promising tool to give insight into the search of good biomarkers useful to discriminate between physiological and pathological aging as well as between age-related and age-associated diseases [19].

Fractal dimension (FD) has been proposed and used as a suitable tool to measure complexity variation of most biomedical functions and structures during aging and disease [48,49,50,51,52,53,54]. While FD gives an estimate of the complexity of the structure, it alone is not sufficient to characterize a fractal object. Fractal lacunarity, another fractal property, instead, describes the texture of a fractal and can measure fractal space filling capacity [55]. The term lacunarity (from Latin lacuna, lack or hole) was coined by Mandelbrot by referring to the gap distribution in a fractal [55], and lacunarity analysis was initially introduced to differentiate fractal objects displaying the same FD but having a very different appearance. Fractal lacunarity analysis has been used to develop our method. In fact, lacunarity analysis has been also introduced as a more general technique able to describe both random and fractal spatial patterns [56,57], thus, overcoming the limits of fractal analysis applied to natural objects [41,55,56]. In addition, dealing with mtDNA mutations as gaps in the nucleotide sequence, fractal lacunarity appears a suitable tool to differentiate between aging and neurodegenerative disease(s).

The present study deals with profiling of PD mutated mtDNA in peripheral blood from PD patients and age-matched controls. CGR method has been used to display DNA fractal properties [58,59,60,61] as modified to visualize heteroplasmic mutations peculiar to mtDNA [33]. Parameter β, from our fractal lacunarity method based on hyperbola model function [62,63,64,65], is the measure used to quantitatively characterize PD on the basis of mutated mtDNA. It is worth noting that lacunarity parameter β, in this case, represents a holistic estimate of mtDNA changes comprehensive of number, type, and dislocation of mutation(s), the combination of which could be responsible of disease onset and progression.

## 2. Results

Thirty mtDNA sequences from peripheral blood of 15 late-onset PD patients and 15 age- and sex-matched controls (Table 1) were processed to generate CGR matrices for L=5 and L=6. Any set of matrices 2^L^x2^L^ was analyzed to verify the potential of our lacunarity parameter β in characterizing alterations of mtDNA in aging and PD, based on the revised Cambridge Reference Sequence (rCRS).

### 2.1. Chaos Game Representation

Figure 1 shows a set of six CGR images generated from rCRS with matrices 2^L^x2^L^ for L=1 to L=6. Note that fractal structure of human mtDNA resembles the Sierpinski triangle (Figure 1b). The same Sierpinski-like structure for human mtDNA was previously reported by Wang et al. [66]. It differs from human nuclear DNA and from DNA of other species, thus highlighting a species- and type-specificity of DNA fractal representation.

Table 2 summarizes results on mtDNA sequences from the whole sample of 30 subjects. In particular, both PD and control mtDNA sequences show decreased numbers of the four nucleotide bases when compared to rCRS. These changes, accounting for similar numbers of no-call, homoplasmic, and heteroplasmic mutations observed in PD and control subjects, do not show statistically significant differences.

### 2.2. Fractal Analysis of mtDNA in Aging and Parkinson’s Disease

Our method of fractal lacunarity analysis was systematically applied to CGR matrices 2^L^x2^L^ generated from the above described mtDNA sequences for L=5 (32x32) and L=6 (64x64) based on results obtained in a previous study on mtDNA in AD **[33].** For the best characterization of any mtDNA sequence analyzed, the combination of coefficients, both sigmoid coefficients (k and σ) and b_min_, was confirmed to be: k equal to 7 and σ equal to 0.7; b_min_ equal to 3 and 5. For this set of coefficients, we obtained comparable results related to rCRS.

Examples of CGR images for L=5 from whole mtDNA sequence of rCRS, a PD patient, and an age-matched control are reported in Figure 2a. In spite of a similar display among the three kinds of mtDNA sequences, parameter β values related to both PD and control subjects significantly differ from rCRS. In particular, lower β values, observed in PD and control mtDNA when compared to rCRS, correspond to the degree of alterations (number, type and dislocation) of the nucleotide sequences considered. In this study, however, we didn’t find any statistically significant difference between PD and control groups for both b_min_ 3 and 5 in matrices 32x32 (L=5) and 64x64 (L=6). Other matrix sizes (i.e., L=4, L=7, L=8) tested didn’t show statistical differences as well. We also noted that two PD subjects and three controls didn’t exhibit the characteristic hyperbola-like lacunarity curve of their mtDNA sequence CGR for L=5 and b_min_=3.

Because of the low number (58) of np differing between PD and control groups [32] we decided to analyze a shorter frame of mtDNA sequence. In particular, we considered the frame length corresponding to 5713–9713 np in the GenBank. It represents about ¼ of the whole mtDNA sequence and contains 41% (24 out of 58) np with increased presence of non-reference alleles in PD patients. Interestingly, mutations in this frame involve genes, such as COI, COII, COIII of complex IV, and ATPase6 of complex V, of the respiratory chain. These complexes play a very important role in energy production; therefore, their mutations may represent a serious impairment of cell metabolism. Figure 2b shows examples of CGR for L=5 of the mtDNA frame considered, and related β values, from rCRS and from the same PD and control subjects of Figure 2a.

Lacunarity analysis of 32x32 CGR matrices generated from this mtDNA frame length showed a significant lower β value in both PD and control groups when compared to rCRS. After exclusion of two PD patients and two controls displaying abnormally high α values, a statistically significant difference was observed between PD patients and age-matched control subjects with lower β values found in controls. Table 3 summarizes lacunarity results from both mtDNA whole and frame lengths for CGR size 32x32 (L=5) and b_min_=3. Randomized sampling method was applied to the sample of subjects under study to confirm that the statistically significant difference between PD and controls is not reached by chance alone. Results from 20-fold repeated randomized sampling into two mixed groups do not show any statistical difference.

## 3. Discussion

In this study we confirm that fractal analysis represents a useful tool to discriminate between aging and age-related pathologies [33,63,65]. In particular, fractal lacunarity analysis of mtDNA, performed by our method, is able to highlight a statistically significant difference of mtDNA mutation profile between PD patients and age-matched controls. Differently from a previous study on AD [33], we found that CGR images of the whole mtDNA sequence from PD patients display fractal properties similar to age-matched controls while parameter β (representative of lacunarity) is significantly lower in both groups when compared to rCRS. Keeping in mind that low β values correspond to high lacunarity, that is high degree of alterations of mtDNA, this result is consistent with previous observations of only 58 statistically different np in PD [32] vs. 270 found in AD [30]. The low number of variants, together with their type and dislocation along the whole mtDNA sequence, is not sufficient to characterize a PD mutation-based profile vs. aging. However, by focusing on a shorter frame of mtDNA sequence, corresponding to GenBank np 5713-9713, that includes 41% of total np whose nucleotide distribution is different between PD and controls, we found a statistically significant difference of lacunarity between PD and control groups with lower β values observed in the last one.

From literature, it is quite accepted that mitochondrial dysfunction and oxidative damage play an important role in the pathogenesis of PD [9,12,18] as it is for other neurodegenerative diseases [27,67,68]. However, the mechanisms involved have not been clearly identified yet. Most mechanisms of mitochondrial dysfunction observed in PD are shared with the aging process [12,69] and overlapping of mtDNA alterations in PD and controls, without a clear-cut characterization of PD mtDNA mutation profile, observed also in this study when dealing with the whole mtDNA sequence, can explain the undefined genotype picture of this disease. As a matter of fact, although more deletions were found in mtDNA of PD patients, comparison with age-matched controls didn’t show any remarkable increase [70,71,72]. With the advent of more sensitive techniques to investigate the deleted mtDNA in individual cells, the increased number of mtDNA deletions in individual neurons of the substantia nigra has been confirmed in PD patients older than 65 years [73]. Another study [74], investigating the number of mtDNA deletions in the brains of subjects with parkinsonism, widespread Lewy body deposition, and less severe loss of dopaminergic neurons in substantia nigra, has reported a high proportion of mtDNA deletions (43%) in controls, which increased with age. However, a higher number of mtDNA deletions was observed in patients with parkinsonism and dementia (52%).

Generally, sequencing of mtDNA from PD patients has been performed in unselected groups, with and without a mitochondrial deficiency [75,76]. Although the results from some studies have suggested increased frequency of specific mtDNA polymorphisms in PD patients, this aspect has not been confirmed in all studies [77,78,79,80,81]. In addition, PD has not been strongly associated with any specific mtDNA mutation yet. The increased frequency of specific mutations or mtDNA haplogroups observed in controls has suggested that they may play a protective role against PD, while some others have been associated with PD [18,69,82,83]. Increased levels of somatic mtDNA point mutations in the substantia nigra have been found also in early PD, suggesting that such mutations occur in the early phase of this pathology [84]. If mtDNA mutations have any effect on this disease, their role appears to be complex and could involve specific haplotypes or combinations of sequence changes that modify mitochondrial function and make the system more vulnerable to nuclear genetic effects and/or environmental influences [12,18].

Such a picture highlights the usefulness of the theory of complexity and the laws of chaos to explain and characterize individual phenotypes evolving over time as ‘normal’ aging or pathological aging [12,33,41]. The inter-individual variability observed in senescent phenotype can be explained in the light of the theory of complexity by considering longevity as a “secondary product” of evolution of a dynamic nonlinear system. In fact, human beings as complex systems are made up of numerous sub-systems (nervous, endocrine, cardiovascular, … systems), interacting with each other. Each sub-system is further subdivided into interacting lower components (organs, tissues, cells, …) and so on at lower levels of organization. The organization into hierarchy and the laws of chaos can explain their evolution, from development to senescence, through the maintenance of a homeostatic dynamic equilibrium of their integrated functions as an adaptive response to continuous noxae from both endogenous and exogenous environments.

Based on this holistic point of view, aging has been defined as the temporal evolution of a complex system that evolves, under the influence of both endogenous and exogenous environments, with loss of complexity during aging [33,45,46,51]. Human beings, as complex systems characterized by a chaotic behavior, generate fractals that can be observed at both structural and functional levels. Fractal analysis, therefore, represents an intriguing tool useful to measure biocomplexity changes with aging and pathology. Fractal analysis can measure variations of complexity in biosystems that evolve with time by following different trajectories. The specific individual genetic-environment interactions determine the senescent phenotype that evolves as ‘normal’ aging, pathological aging, or successful aging [33,40]. Dealing with mutations of mtDNA as gaps in the nucleotide sequence, fractal lacunarity represents the most suitable tool to differentiate between PD and aging. In fact, lacunarity gives a holistic estimate of changes that occur in mtDNA sequences, comprising of number, type, and dislocation of mutation(s), the combination of which could contribute to PD onset and progression.

It is worth noting that a full certain diagnosis of PD is impossible during life: 75%–95% of PD patients have their diagnosis confirmed only post-mortem [85]. Diagnostic accuracy varies considerably depending on disease duration (lower on first visit), age, clinician expertise, and increasing understanding of this pathology. In this context, advances in live imaging, such as neuromolecular imaging, could improve the understanding as well as diagnosis of PD [31,86]. Often, failure in recognizing other pathologies causing neurodegenerative disorder or secondary parkinsonism as well as the absence of a true progressive parkinsonian disorder are common causes of diagnostic error [85].

Dealing with this and other observations discussed above, it would be interesting to verify the potential of our original method for mtDNA mutation-based profiling of neurodegenerative disease(s) also for the whole mtDNA sequence from PD patients as per AD [33]. In fact, we don’t know whether the positive results observed on the smaller mtDNA frame but not on the whole sequence is peculiar to the disease or, rather, it is attributable to the sample of subjects used in this study. As a matter of fact, PD patients were diagnosed for late-onset PD; therefore, accumulation of mtDNA somatic mutations that occur with aging could mask PD specific point mutations or sequence changes of mtDNA. In addition, taking into account that PD is a neurodegenerative disease characterized by motor dysfunction, it would be useful to compare PD with age-matched controls without altered motor function. Our sample didn’t show any statistically significant difference as far as cognitive and motor function, unrelated to PD dysfunction, are concerned, the only difference accounting for limited head rotation in PD group. Last but not least characteristic of the sample under study deals with hospitalization: both PD patients and age-matched controls were hospitalized for other and different pathologies, probably with or without implication for mitochondrial dysfunction. In this context, it would be interesting to investigate whether the similar mtDNA mutation profile, observed in the whole mtDNA sequence of both PD and controls, could represent a mitochondrial genome profile peculiar to pathological senescent phenotype, characterized by multimorbidity affecting more than 60% elderly people that experience a pathological aging.

## 4. Materials and Methods

### 4.1. Subjects

This study was performed on mtDNA sequences from a previous study [32] in which blood samples were obtained from the institutional Report-AGE Database Sample Resource of IRCCS INRCA (approved by the Institute Bioethics Committee - 12/DSAN; 19 April 2011) [87]. In this database, data and biological samples are preserved from patients that had previously given written informed consent in accordance with the Declaration of Helsinki. The blood samples came from 15 PD patients and 15 control subjects matched for both age and sex. PD patients included in the study were diagnosed according to Postuma et al. [85], treated with levodopa, with late disease onset (65 years and older) and no family history of PD.

Exclusion criteria for PD patients were a history of other neurological disorders, a diagnosis of PD as a secondary disease, and PD with atypical disease progression. Exclusion criteria for control subjects were a history of any neurological disease, orthostatic hypotension, cognitive impairment, and parkinsonism. All participants were of Caucasian ethnicity and underwent cognitive status and motor function assessment [88]. Table 1 summarizes the characteristics of the study population.

### 4.2. mtDNA Extraction and Resequencing

MitoChips were prepared as previously described [30]. In particular, data set was acquired by the Affymetrix GeneChip Command Console software and analyzed with the GeneChip Sequence Analysis software 4.1 (Affymetrix, Santa Clara,CA). This software elaborates fluorescence intensity data by means of an algorithm whose parameters were defined to achieve optimal performance in analyzing mitochondrial sequences, with “model type” set at diploid to enable the detection of heteroplasmy and “quality score threshold” set at 3 to provide the best base calling accuracy and rate. We included in the analysis chips whose call rate was >95% implying that unclassified np, known as no-call, had to represent a very small percentage of total calls. The output files used for this study were the Single Nucleotide Polymorphism View files. These files provide the base calls, including the homoplasmic and heteroplasmic variants, through comparison with rCRS.

Mitochip sequencing has been extensively used to study mtDNA mutations associated with cancer, AD, polymorphisms, and rare mutations [89,90,91,92]. Mitochip sequencing has been well characterized in terms of sensitivity, specificity, and accuracy in detecting homoplasmic and heteroplasmic variants, and has been demonstrated to provide reproducible results, thus making a verification procedure unnecessary [93,94]. Mitochip sequencing and Next Generation Sequencing show comparable performances as regards base call accuracy. However, the former technique is easier to use and involves less labor-intensive sample preparation.

Data discussed here were deposited in NCBI’s Gene Expression Omnibus [95] and are accessible through GEO Series accession number GSE113704 (https://www.ncbi.nlm.nih.gov/geo/query/acc.cgi?acc=GSE113704).

### 4.3. Chaos Game Representation of mtDNA

The CGR method as modified in Zaia et al. [33] was used to analyze the structure of mtDNA from Affymetrix MitoChips. Roughly speaking, CGR from the pioneering work by Jeffrey [58] allows the codification in images of the information contained in long sequences of symbols; therefore, it can be also applied to DNA to analyze this information by the fractal features of the corresponding CGR images.

Let *S* be a finite string given by a DNA sequence, where the alphabet is {a, c, g, t}. The CGR image of *S* is obtained by the frequency occurrence of all the possible substrings of *S* having fixed length *L*. Due to the four-symbol alphabet of S, this map can be organized in a square matrix of order 2*^L^*. Figure 3 shows a pictorial description of this organization for *L=1,2,3*.

A formal definition of the structure of matrix *M_L_*, in the generic case *L*, can be given by matrix tensor product of 2x2 matrices *M=M_1_* in the case *L*=*1*; more precisely:(1)$$M_{L}=\underset{L\;\mathrm{times}}{\underset{︸}{M⊗M⊗\cdots ⊗M}}$$

From this relation, a position *p(s)* is associated to each possible substring *s*⸦*S* of length *L*. Thus, the following algorithm can be used to construct the CGR matrix:

**Algorithm** **1.***Let L be a positive integer and S be a DNA sequence; construct the CGR matrix A of S by the following steps*:
initialize A to a zero matrix having 2^L^ rows and 2^L^ columns,for each substring s⸦S, increase by one the entry of A having position p(s).

An efficient computer implementation of this algorithm is described in Vinga et al. [96].

We proposed a slight modification of Algorithm 1 in order to deal with undetermined DNA typing symbols (heteroplasmic mutations) [33]. More precisely, substrings *s* containing undetermined symbols are substituted by the corresponding multiple strings obtained by solving such symbols and each one of such strings has a fractional weight *w* computed by the number of generated strings.

The following examples show the generation procedure for the strings and the weights.

**Example** **1.**
*Let L=5 and s=’tamcg’, where the undetermined symbol ‘m’ means ‘a’ or ‘c’. This string is substituted by s_1_=’taacg’ and s_2_=’taccg’; the weight is w=1/2.*


**Example** **2.**
*Let L=6 and s=’tavcgm’, where ‘v’ means ‘a’, ‘c’, or ‘g’. This string is substituted by s_1_=’taacga’, s_2_=’taccga’, s_3_=’tagcga’, s_4_=’taacgc’, s_5_=’taccgc’, s_6_=’tagcgc’; the weight is w=1/6.*


In particular, this generation procedure is inserted in Algorithm 1; in this way, the resulting algorithm is able to work also with substrings containing undetermined symbols.

**Algorithm** **2.**
*Let L be a positive integer and S be a DNA sequence; construct the CGR matrix A of S by the following steps:*

*initialize A to a zero matrix having 2^L^ rows and 2^L^ columns,*

*for each substring s⸦S, compute the number N of multiple strings generated by the undetermined symbols in s, and perform the following cycle,*

*for n=1,2,…,N,*
i.
*generate the string s_n_ from s,*
ii.
*increase by w=1/N the entry of A having position p(s_n_).*


*end for.*



This algorithm has been implemented in a MATLAB program (the MatWorks, Inc.) that, also, provides a report with additional information on mtDNA sequence processed, i.e., type and number of nucleotide(s), number and position of homoplasmic/heteroplasmic mutations.

### 4.4. Estimate of Lacunarity

Fractal lacunarity analysis of CGR matrix has been performed by using the gliding box algorithm (GBA), see [57] for a detailed description. This method has been applied to several medical investigations and it is already described in [62,63] and in [64,65] for a modified version. In this section, GBA is sketched out for the convenience of the reader.

GBA is based on the analysis of the mass distribution in a given set, where the set is associated to the CGR matrix and the mass is associated to the total frequency obtained in the elements of CGR matrix. In particular, GBA involves the moments of the box mass, M, within a box moving on the set one space unit at a time. The computation of the box mass is repeated for all the different boxes traversing the set. In this way, a frequency distribution n(M,b) of box masses is obtained, where b is the size of the gliding box.

For each b, let Mj, j=1,2,…,μ(b) be the different masses encountered in the various gliding boxes of size b; so, a discrete frequency distribution n(Mj,b), j=1,2,…, μ(b) has to be considered. From standard arguments on probability, the moments of order q of M, are given by:(2)$$Z_{q}(M,b)=\frac{1}{N(b)}\sum_{j=1}^{\mu (b)}M_{j}^{q}n(M_{j},b),\;b>0,$$
where N(b), i.e., the total number of boxes, needs to convert n(Mj,b), j=1,2,…,μ(b) into a probability distribution. The definition of lacunarity function Λ uses only the first and the second moments of M, that is
(3)$$\Lambda (b)=\frac{Z_{2}(M,b)}{Z_{1}{\left(M,b\right)}^{2}},\;b>0$$

This algorithm can be used to deal with gray scale images and provides a simple extension of the method for binary images, see Zaia et al. [64,65] for details. The efficiency of such an algorithm is usually improved by a pre-processing, where the original image I is used to compute a refined version J as follows:(4)$$J(i,j)=\frac{1}{1+\exp(-k(I(i,j)-\sigma ))},\;i=1,2,\ldots ,Nrows,\;j=1,2,\ldots ,Ncolumns,$$
and k, σ > 0 are two given parameters. It is worth noting that the procedure goes toward a complete binarization by increasing parameter k, related to sigmoid regularization.

The GBA method has been implemented in a Matlab program that calculates the values of lacunarity Λ(b), for each integer value of b between b_min_ and b_max_, where b_min_, b_max_ are given integer multiples of the pixel size in the image under consideration. Once the lacunarity function Λ(b), b=b_min_, b_min_+1,…, b_max_ is obtained, the program shows the results on a graph and computes the least squared best-fit through the data by using the following model:(5)$$L(b;\alpha ,\beta ,\gamma )=\frac{\beta }{b^{\alpha }}+\gamma \;b\in [b_{\min},b_{\max}]$$
where α, β, γ are suitable parameters. So, the fractal properties of the CGR are described by the best-fit parameters (α*, β*, γ*) computed as the minimizer of this optimization problem:(6)$$\underset{\alpha ,\beta ,\gamma >0}{\min}\sum_{b=b_{\min}}^{b_{\max}}(\Lambda (b)-L(b;\alpha ,\beta ,\gamma ){)}^{2}.$$

We note that parameter α is related to the fractal dimension of the set and parameter β characterizes the lacunarity of the set [62,63].

Figure 4 shows a schematic representation of the method applied to a CGR matrix generated by rCRS. CGR matrices generated from different mtDNA sequences produced a similar curvilinear plot. The almost perfect overlap of the two experimental and theoretic curves supports the appropriateness of our choice of hyperbola model function to fit the gliding box curve.

### 4.5. Statistical Analysis

All analyses were performed using SPSS for Windows, version 19.0. All data with normal distribution were presented as means ± SD. Student’s t-test was used to compare differences between PD and control groups. Data without normal distribution were expressed as median (interquartile range) and analyzed by Mann–Whitney U Test. Statistical significance was accepted for *p*≤0.05.

## 5. Conclusions

In conclusion, our method of lacunarity analysis on CGR from mtDNA represents a useful tool to analyze mtDNA mutation profile.

Parameter β from our lacunarity method, using hyperbola model function, is able to characterize the individual mutation profile of mitochondrial genome and appears to be a promising index to discriminate between PD and aging.

Results presented in this paper are from a pilot study on late-onset PD and age-matched controls coming from a hospitalized cohort of individuals. As such, they need to be confirmed in larger and different samples of subjects.

Biocomplexity, chaos, and fractality can provide a promising approach to give insight into the search of biomarkers of aging and pathologies.

## Figures and Tables

**Figure 1 ijms-21-01758-f001:**
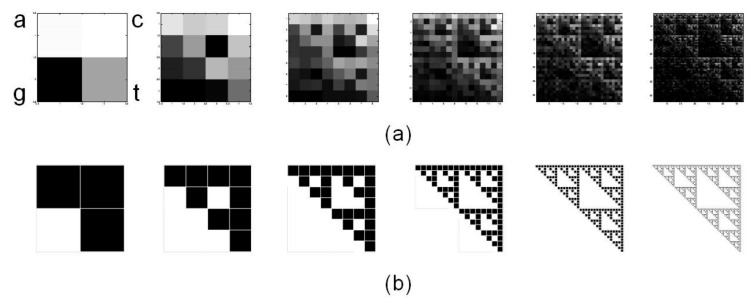
Chaos Game Representation (CGR) of human mtDNA. (**a**) CGR matrices for L=1 to L=6 of whole revised Cambridge Reference Sequence, rCRS, display self-similarity, a property peculiar to fractals, resembling (**b**) the triangle of Sierpinski, an ideal fractal built through repeated iterations starting from a square.

**Figure 2 ijms-21-01758-f002:**
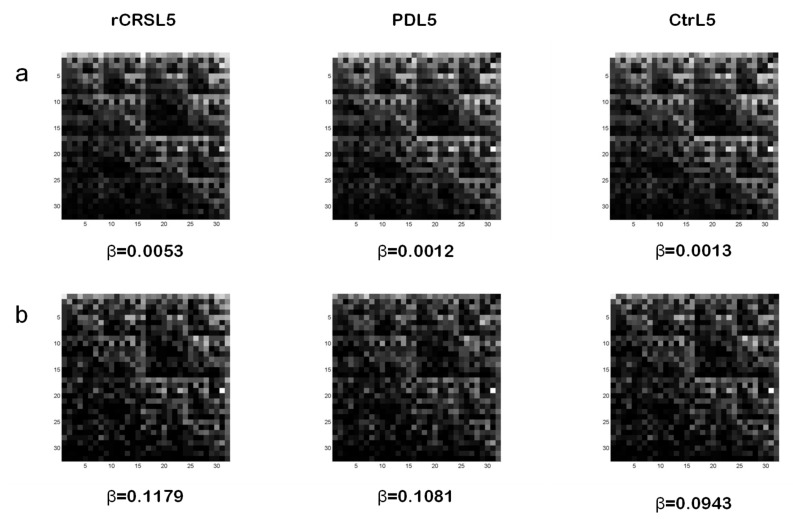
Examples of CGR images of whole mtDNA sequences (**a**) and related GenBank np 5713-9713 frames (**b**) from different subjects. Lacunarity parameter β value for CGR matrices 32x32 (L=5) generated from mtDNA of rCRS (left), a PD patient (middle), and a control (right) is reported. CGR: Chaos Game Representation; rCRS: revised Cambridge Reference Sequence; PD: Parkinson’s Disease.

**Figure 3 ijms-21-01758-f003:**
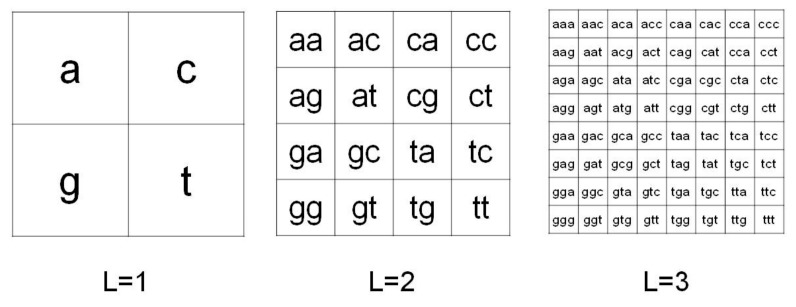
Chaos Game Representation method. CGR organization in matrices 2^L^x2^L^ for L=1,2,3 in the case of four-symbol alphabet {a,c,g,t}.

**Figure 4 ijms-21-01758-f004:**
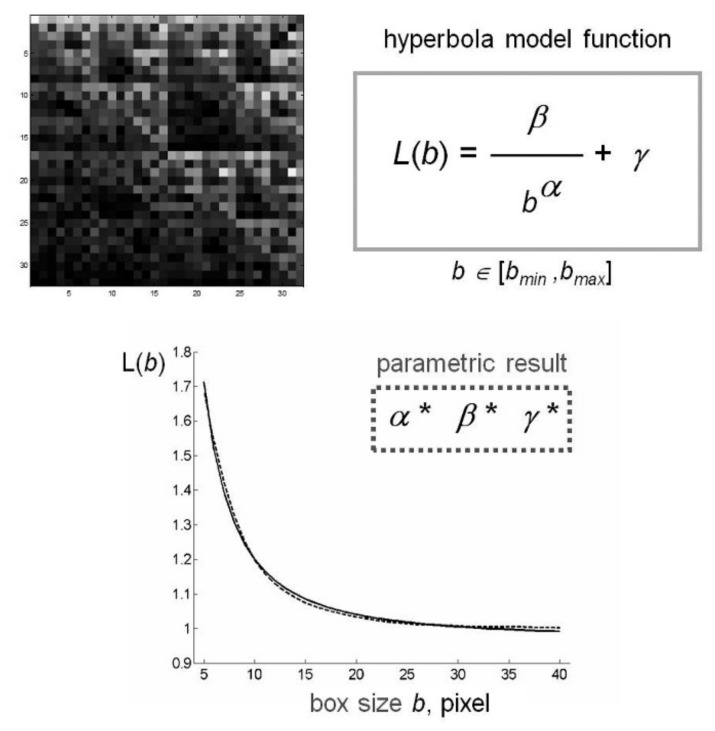
Schematic representation of fractal lacunarity analysis. (Top left) rCRS mtDNA image generated by CGR matrix for L=5 is a 32x32 square. The plot (bottom) represents the result of GBA application (dotted line), for bmin=3, as fitted by hyperbola function (solid line) used to calculate the triplet of parameters α, β, γ. CGR: Chaos Game Representation; rCRS: revised Cambridge Reference Sequence; GBA: Gliding Box Algorithm.

**Table 1 ijms-21-01758-t001:** Characteristics of subjects included in the study.

	PD Patients	Controls
Number (M/F)	15 (7/8)	15 (6/9)
Age (year)	78.8 ± 6.0	80.8 ± 5.1
H&Y	2.5 (2.0-3.0)	
CPS	2.0 (0.5-3.0)	2.0 (0.5-2.0)
ADL	1.0 (0.0-2.0)	0.0 (0.0-3.0)

Values are expressed as mean ± SD or median (interquartile range) in the case of non-normally distributed variables; PD: Parkinson’s Disease; H&Y: Hoehn & Yahr stage; CPS: Cognitive Performance Scale; ADL: Activities of Daily Living hierarchy scale.

**Table 2 ijms-21-01758-t002:** Characteristics of mtDNA sequences processed by the proposed method.

Number	rCRS	PD patients	Controls	p Value
Subjects		15	15	
Adenine	5117	4884 ± 252	4912 ± 110	0.359
Cytosine	5175	4658 ± 206	4618 ± 168	0.301
Guanine	2163	2102 ± 92	2108 ± 49	0.421
Thymine	4089	3880 ± 239	3916 ± 101	0.311
No-call	-	1009 ± 765	980 ± 390	0.453
Homoplasmy	-	18 ± 8	21 ± 8	0.196
Heteroplasmy	-	12 ± 13	10 ± 6	0.332

Values are expressed as mean ± SD; p Values have been calculated by one-tailed t-test for p≤0.05 to compare differences between PD and control groups; rCRS: revised Cambridge Reference Sequence; PD: Parkinson’s Disease.

**Table 3 ijms-21-01758-t003:** Fractal parameters from lacunarity analysis method based on hyperbola model function of mtDNA in Parkinson’s disease.

mtDNA	whole		frame^1^	
	α	β	α	β
**rCRS**	**1.6375**	0.0053	1.3224	0.1179
PD patients	1.6405 ± 0.0005	0.0011 ± 0.0004	1.3106 ± 0.060	0.1019 ± 0.061
Controls	1.6403 ± 0.0007	0.0009 ± 0.0013	1.3150 ± 0.073	0.0910 ± 0.061
p Value	0.179	0.151	0.433	0.028

^1^ mtDNA sequence frame corresponding to GenBank np 5713-9713.

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
