# Peer review of "Biocomplexity and Fractality in the Search of Biomarkers of Aging and Pathology: Mitochondrial DNA Profiling of Parkinson’s Disease"

_ijms, 2020, doi:10.3390/ijms21051758_

Round 1

Reviewer 1 Report

 It is highly interesting to use fractal analysis to discriminate between aging and age-related pathologies. The author developed a method which is a useful tool to analysis mtDNA muattations and represent a promising index to discriminate PD and aging. I suggest this manuscript is suitable to publish.

Author Response

We are very glad the reviewer has appreciated our work.

Reviewer 2 Report

Zaia et al have studied mitochondrial DNA sequences to determine whether there are specific mutation profiles that can used to distinguish between aged from Parkinson’s disease cohorts. Overall, the approach is interesting but limited its findings.

Major concerns:

1) To reach statistical significance, the authors have focused on a region of mtDNA (5713-9713 np) and eliminated samples from two PD patients (p5 line 169). This is troublesome as it suggests that the authors are filtering and fitting the data to search for a significant result. It becomes unclear whether there is true significance owing to the already low number of samples used. Could the authors perform statistical analyses that would give greater credibility to their overall conclusions? E.g. if the 30 samples are mixed and then randomly distributed into two different groups, would statistically significant patterns be found between groups? If so, then the study is under powered. This randomised experiment would need to be repeated many different times to demonstrate that the conclusions of the paper are not reached by chance alone. As it stands, the main claim of the manuscript is not so well supported by the data, and needs further analysis.

2) I also have a concern about the methods used to obtain mtDNA sequence data. MitoChips, a rather outdated method, were used and not sequencing. This method will not derive actual nt changes. This decreases the accuracy of the analyses. Also, how many technical replicates for each were performed is not clear.

Minor concerns:

Line 26: “useful tool to analyse mtDNA mutations” In which sense? The exact nt change cannot be derived from the techniques. The model only displays abberations without giving further information.

Line 34: over65s/over85s is an unusual way of phrasing age. Replace with over 65s.

Line 40: refs 2-7 are 18-23 years old. Are there any more recent studies on this?

Line 40-42: “At present, the common opinion is…” reference is from 1983.

Line 56/57: If there are organ specific effects of heteroplasmy, which is true, why would it be possible to conclude that changes in mtDNA sequence in the blood could predict brain changes?

Line 57: More up to date references are needed throughout the manuscript. Here you could cite some recent papers expanding on heteroplasmy e.g. Hahn et al 2019 Trends Cell Biol 29 p227.

Line 63: Please specify what an adaptive polymorphism is.

Line 62-64: Please clarify.

Line 71: define ROS.

Line 91: change to pathological aging.

Line 93: What is the difference between age-related and age-associated disease?

Line 109: Parameter Beta indicates the mutation load in PD patients? This is important and needs more explanation.

Line 132: This sentence is not well written or phrased.

Line 133: This text does not describe the table very well.

Line 153: What is the consequence of that and why could this be?

Line 161: Why is this mtDNA region significant? Any reason?

There are also multiple typos and grammatical mistakes.

Author Response

Please, see attachment.

Reviewer 3 Report

Recommendation: Publish with minor revisions.

Lacunarity is an important advance. We need new approaches to Parkinson's disease. We need to distinguish aging from Parkinson's disorders as well as distinguishing aging from neurodegeneration of brain processes that come with inherent dementias as well. We need to bring chaos and entropy into the equation.

From the authors':
"Biocomplexity, chaos and fractality can provide a promising approach to give insight into the search of biomarkers of aging and pathologies" True!

But, the limitation is, as the authors say, the "N" number is too small. This is acceptable for an emerging technology especially using mitochondrial DNA. 

Moreover, we have some suggestions for references that the authors' will like to hear and reference in their manuscript. The authors say, a priori, that there are many ways to study devastating brain diseases. 

These other ways deserve citation. We would like to share these other ways. LIVE IMAGING studies in the Parkinson's brain, on line and in real time are part and parcel of new advances. New molecules appear in the Parkinson's brain within sub seconds and before our eyes using Neuromolecular Imaging (NMI). We have given two references for these cutting edge NMI studies. The second reference brings the etiology of Parkinson's to the gastrointestinal tract via beta amyloid.

We hope the authors' enjoy our suggestions and revise their manuscript accordingly. 

The Reference List for perusal and citation is attached to this email in the Authors' section for comments and suggestions. 

REFERENCE LIST: 

Broderick PA., Wenning L., “Neuromolecular imaging in Parkinson's disease”. In: (Ed. Victor R. Preedy) Compendium on Parkinson’s disease. Chapter 16. (2020) Elsevier Press, London, UK.

Borghammera P., Van Den Bergeb N. “Brain-First versus Gut-First Parkinson’s Disease: A Hypothesis”. Journal of Parkinson’s Disease (2019) 9: S281–S295.

Author Response

We are glad to the Reviewer 3 for sharing  with us the two references on neuromolecular imaging and gastrointestinal Parkinson’s.

We revised the manuscript according to Reviewer 3 by adding the suggested references. In particular, Borghammera P, Van Den Bergeb N, 2019 have been added with additional text in Introduction, line 75-76, reference [32].

The second reference (Broderick PA, Wenning L, 2020) , together with the first one, has been cited [88] with additional text in Discussion, line 266-267.